# Interaction of Cationic Carbosilane Dendrimers and Their siRNA Complexes with MCF-7 Cells Cultured in 3D Spheroids

**DOI:** 10.3390/cells11101697

**Published:** 2022-05-19

**Authors:** Kamila Białkowska, Piotr Komorowski, Rafael Gomez-Ramirez, Francisco Javier de la Mata, Maria Bryszewska, Katarzyna Miłowska

**Affiliations:** 1Department of General Biophysics, Faculty of Biology and Environmental Protection, University of Lodz, 141/143 Pomorska St., 90-236 Lodz, Poland; maria.bryszewska@biol.uni.lodz.pl (M.B.); katarzyna.milowska@biol.uni.lodz.pl (K.M.); 2Molecular and Nanostructural Biophysics Laboratory, “Bionanopark” Ldt., 114/116 Dubois St., 93-465 Lodz, Poland; p.komorowski@bionanopark.pl; 3Department of Biophysics, Institute of Materials Science, Lodz University of Technology, 1/15 Stefanowskiego St., 90-924 Lodz, Poland; 4Department of Organic and Inorganic Chemistry, IQAR, University of Alcalá, 28805 Madrid, Spain; rafael.gomez@uah.es (R.G.-R.); javier.delamata@uah.es (F.J.d.l.M.); 5Networking Research Center on Bioengineering, Biomaterials and Nanomedicine (CIBER-BBN), 28029 Madrid, Spain; 6Ramón y Cajal Health Research Institute (IRYCIS), 28034 Madrid, Spain

**Keywords:** siRNA, carbosilane dendrimers, dendriplexes, nanocarriers, 3D cell culture, spheroids

## Abstract

Cationic dendrimers are effective carriers for the delivery of siRNA into cells; they can penetrate cell membranes and protect nucleic acids against RNase degradation. Two types of dendrimers (CBD-1 and CBD-2) and their complexes with pro-apoptotic siRNA (Mcl-1 and Bcl-2) were tested on MCF-7 cells cultured as spheroids. Cytotoxicity of dendrimers and dendriplexes was measured using the live–dead test and Annexin V-FITC Apoptosis Detection Kit (flow cytometry). Uptake of dendriplexes was examined using flow cytometry and confocal microscopy. The live–dead test showed that for cells in 3D, CBD-2 is more toxic than CBD-1, contrasting with the data for 2D cultures. Attaching siRNA to a dendrimer molecule did not lead to increased cytotoxic effect in cells, either after 24 or 48 h. Measurements of apoptosis did not show a high increase in the level of the apoptosis marker after 24 h exposure of spheroids to CBD-2 and its dendriplexes. Measurements of the internalization of dendriplexes and microscopy images confirmed that the dendriplexes were transported into cells of the spheroids. Flow cytometry analysis of internalization indicated that CBD-2 transported siRNAs more effectively than CBD-1. Cytotoxic effects were visible after incubation with 3 doses of complexes for CBD-1 and both siRNAs.

## 1. Introduction

Gene therapy is one of the most promising and effective ways of treating cancers, this disease being a major cause of death in the world [1,2,3,4,5,6]. Gene therapy is the ability to make genetic improvements by correction of mutated genes or site-specific modifications for therapeutic purposes [7], and it results in the regulation or replacement of altered genes when therapeutic genes are transported into the chromosomes of target tissues or cells [8], with genes being used as medicines [9]. Correction of defective genes is performed in several ways: (1) The nonfunctional gene can be replaced by a normal gene, inserted within the genome into a nonspecific location; (2) A mutated gene can be exchanged for a recombined homologous or normal gene; (3) A mutated gene can be repaired by using selective reverse mutation; (4) The degree to which a gene is turned on or off can be altered [9]. Target diseases for gene therapy include those caused by recessive gene disorders, e.g., hemophilia, cystic fibrosis, sickle-cell anemia and muscular dystrophy, and those that are acquired, e.g., cancers. Target diseases also include some viral infections such as AIDS [7].

A cell membrane is a strong barrier against genes in the form of large and anionic nucleic acids, DNA or RNA [3]. For this reason, effective application of gene therapy is dependent on technologies that allow delivery of genes into cells, tissues and organs [10]. It means that the most important factor determining the implementation of gene therapy is the development of effective vectors that are also safe for healthy organs [10,11]. Two types of vectors are currently available as gene carriers: viral or non-viral [11]. With viral vectors, most of the viral genome is replaced by the therapeutic nucleic acids. Transport into the cells is possible due to viral proteins that mediate internalization and prevent degradation of the vector in the extracellular environment [3]. Viral vectors provide a possibility of efficient transfection and long-term silencing of gene expression. However, there are some problems with using viral vectors as gene carriers, e.g., high cost of production and more importantly, the generation of various side effects and induction of immune responses that lead to a reduction of the therapeutic effect. For these reasons, attention has been concentrated on new non-viral carriers [5,12].

The most commonly tested non-viral vectors include nanoparticles, e.g., liposomes, carbon nanotubes and dendrimers [5]. Dendrimers, which are synthetic polymers, are promising because of their size and water solubility [13]. Cationic dendrimers can bind with nucleic acids and pass through the cell membrane due to their positive charge [4,14]. Other advantages of dendrimers include a large number of functional groups for nucleic-acid binding and their stability [13,14,15]. Therefore, we examined two types of cationic, carbosilane dendrimers: CBD-1 and CBD-2 (see Figure 1 in Section 2), described previously [14] as carriers for two types of small interfering RNAs (siRNAs): Mcl-1 and Bcl-2.

siRNA is a small, double-stranded nucleic acid [16], which can promote messenger RNA (mRNA) degradation [17,18]. This process is called RNA interference (RNAi) and results in silencing of target genes [17,18]. The RNAi process with the use of Mcl-1 and Bcl-2 can be applied in anticancer therapy, since the proteins encoded by these siRNAs are involved in regulation of apoptosis [19,20]. Application of pro-apoptotic Mcl-1 and Bcl-2 can lead to neutralization of mRNA encoding anti-apoptotic proteins and this, finally, can lead to induction of apoptosis in cancer cells [14,17,18,21].

Dendriplexes formed by CBD-1 and CBD-2 with both siRNAs were characterized in our previous work [14]. We also carried out preliminary research with MCF-7 cells, testing cytotoxicity of dendrimers and dendriplexes. However, those experiments used 2D cell culture. 2D models limit cell–cell and cell–matrix interactions and significantly reduce cellular responses, and there is no normal physiological process such as transport of nanoparticles through cell layers [22,23]. Naturally, cells reside in the 3D environment, which is crucial for their growth and metabolism. The functions and the phenotype of each cell depend on complex interactions with extracellular matrix (ECM) proteins and neighboring cells [6,22]. Therefore, we investigated dendrimers and dendriplexes with MCF-7 cells cultured as 3D spheroids and examined cytotoxicity of the tested materials and their uptake by cells. Articles presenting the influence of dendrimers as siRNA carriers on cells cultured in 2D are common, but it is harder to find articles presenting the impact of these systems on cells cultured in 3D.

The data show that both dendrimers are taken up by the cells and there is a difference between cytotoxicity of CBD-1 and CBD-2. After 24 h of treatment with CBD/siRNA, the viability of cells did not decrease when compared to viability after 24 h of treatment with dendrimer alone. However, analysis of cytotoxicity after incubation with 3 doses of dendriplexes revealed the toxic potential of CBD-1 complexed with both siRNAs.

## 2. Materials and Methods

### 2.1. Carbosilane Dendrimers

The second generation carbosilane dendrimers, CBD-1 and CBD-2, were synthesized and characterized as described previously [14,24]. The molecular structure of investigated dendrimers is shown in Figure 1.

### 2.2. siRNA

Two types of anticancer siRNAs used in this study (Mcl-1 and Bcl-2) were described previously [14].

Mcl-1,

Sense Mcl-1s:5′-GGACUUUUAUA-CCUGUUAUdTdT-3′

Antisense Mcl-1a:5′-AUAACAG-GUAUAAAAGUCCdTdT-3′

Bcl-2,

Sense E1s:5′-G CUG CAC CUG ACG CCC UUCdTdT-3′

Antisense E1a:5′-GAA GGG CGU CAG GUG CAG CdTdT-3′

siRNAs and siRNAs-FITC were obtained from Dharmacon Inc. (Lafayette, CO, USA).

### 2.3. Other Reagents

Reagents for cell culture were provided by Biowest (Riverside, MO, USA). TrypLE for cell detachment was from Gibco (Amarillo, TX, USA). Dimethyl sulfoxide (DMSO), phosphate buffered saline (PBS) tablets, 4-(2-hydroxyethyl)-1-piperazineethanesulfonic acid (HEPES) and formaldehyde were purchased from Sigma-Aldrich (St. Louis, MO, USA). Hoechst 33342 was obtained from Santa Cruz Biotechnology, Inc. (Dallas, TX, USA). Calcein AM and propidium iodide (PI) were from Biotium (San Francisco, CA, USA). Annexin V-FITC Apoptosis Detection Kit was from BD Pharmingen (Franklin Lakes, NJ, USA). Tris was obtained from GE Healthcare (Chicago, IL, USA). Other basic chemical reagents (sodium dihydrogen phosphate, disodium hydrogen phosphate, acetic acid, sodium chloride, glucose, potassium chloride, sodium bicarbonate, disodium hydrogen phosphate, potassium dihydrogen phosphate, magnesium chloride 6 hydrate, magnesium sulfate 7 hydrate) were obtained from the local supplier, Chempur, (Piekary Śląskie, Poland). Agarose was purchased from the local supplier, Blirt (Gdańsk, Poland). These chemicals were of analytical grade, and solutions were prepared using water purified using the Mili-Q system.

### 2.4. Dendrimer/siRNA Complexes Formation

Dendrimers and siRNAs were mixed in phosphate buffer (pH = 7.4) at concentrations with molar ratio 20:1 where complexes formed immediately. The complexes were characterized previously [14].

### 2.5. MCF-7 Cell Line

The MCF-7 cell line used in this study was purchased from American Type Culture Collection (ATCC^®^, cat. No: HTB-22, Manassas, VA, USA). The cells were cultured in Dulbecco’s modified Eagle’s medium (DMEM) supplemented with 10% fetal bovine serum (FBS) and antibiotics. Cultures were maintained in an incubator with 5% CO_2_, at 37 °C.

In 2D cell cultures, the cells were seeded onto 12-well plates at the density of 6 × 10^5^ cells per well. Dendrimers or dendriplexes were added after 24 h.

For preparing 3D cell culture, the agarose gels produced using 3D Petri Dish^®^ for small spheroids from MicroTissues Inc. were used (Providence RI, USA). Hydrogel was prepared according to the manufacturer’s instruction using 12-well plates. 2% agarose in PBS was placed in micro-mold and allowed to dry, then the gels were transferred into 12-well plates and the cells were seeded at a density of 1 × 10^5^ per agarose gel and cultured for 7 days in conditions described above. Microscopy image of spheroid after 7 days of culture is shown on Figure 2. After 7 days dendrimers or dendriplexes were added. The diameter of spheroids on the measurement day was <350 nm.

### 2.6. Flow Cytometry Assays

For cytometry analysis of the cells cultured in 2D, the media from each well were transferred into centrifuge tubes and centrifuged for 5 min. at 300× *g* to recover dead cells, after centrifugation the supernatant was discarded. The cells on the plates were washed with PBS and then detached with TrypLE for approx. 5 min. in the incubator with 5% CO_2_, at 37 °C. Hanks’ balanced salt solution (HBSS) with 10% FBS was added. The cells from the plates were transferred to the dead cells in the tubes and mixed. All cells were transferred to the cytometry tubes through cytometry filters and stained with fluorescent dyes. 10,000 events were collected from each sample.

For analysis, the 3D cell cultures after incubation with dendrimers or dendriplexes, were washed with PBS and the cells were removed from gels according to manufacturer’s protocol. TrypLE was added (0.5 mL) onto the new 12-well plate, the gels were transferred to this plate and inverted upside down without any bubbles. The plates were centrifuged for 5 min., at 500 rpm, gels were removed and the cells were incubated for 10 min. in the incubator with 5% CO_2_, at 37 °C. After incubation, 0.25 mL of the HBSS with 10% FBS were added, the cells were gently mixed with automatic pipette, transferred into the cytometry tubes through cytometry filters and stained with fluorescent dyes. 10,000 events were collected from each sample.

#### 2.6.1. Live–Dead Assay

Cytotoxicity effect of dendrimers and dendriplexes was evaluated using the live–dead test. The cells in 2D cell culture were incubated with dendrimers (at 0.5, 1 and 2 μM). The cells in 3D cell culture were incubated with dendrimers (at 5, 10, 20, 30 and 50 μM) or dendriplexes (dendrimer concentration of 5 and 10 μM; dendrimer/siRNA molar ratio 20:1) for 24 or 48 h. Untreated cells were used as a negative control (NC), and the cells treated with 1 mM (2D) or 3 mM H_2_O_2_ (3D) were used as a positive control (PC). After incubation, the cells, prepared as described above, were stained with calcein AM (labeling of live cells) and with PI (labeling of dead cells) and analyzed using flow cytometer LSR^®^ II (Becton Dickinson, Erembodegem, Belgium).

The live–dead assay was also performed after treatment with 3 doses of dendrimers and dendriplexes with cells being cultured as described above. After 7 days of culture the first dose of dendrimers or dendriplexes was added. The second dose was added after 48 h and third dose was added after the next 48 h. After treatment the cells were prepared as described above and analyzed using flow cytometer LSR^®^ II (Becton Dickinson, Erembodegem, Belgium).

#### 2.6.2. Apoptosis Detection by Flow Cytometry (Annexin V-FITC Apoptosis Detection Test)

After 24 h incubation with dendrimers or dendriplexes, the cells cultured in 3D were stained using the Annexin V-FITC Apoptosis Detection Kit. The cells were washed 3 times with PBS, then the cell suspension was prepared as described above. The suspension was centrifuged at 130× *g* for 5 min., resuspended in the binding buffer and stained with the Annexin V-FITC Apoptosis Detection Kit according to manufacturer’s instructions. Finally, the cells were analyzed using the flow cytometer LSR^®^ II (Becton Dickinson, Erembodegem, Belgium).

#### 2.6.3. Cellular Uptake

To estimate the cellular uptake of dendriplexes, complexes of dendrimers with fluorescein (FITC) labeled siRNA were used. The cells were treated with dendriplexes for 24 h (dendrimers at 10 μM; dendrimer/siRNA molar ratio 20:1). For cytometry analysis the cells were prepared as described above and analyzed using flow cytometer LSR^®^ II (Becton Dickinson, Erembodegem, Belgium).

### 2.7. Confocal Analysis of Cellular Uptake

Cells cultured in 3D were used for confocal analysis. After incubation with dendrimer/FITC-labeled siRNA complexes (dendrimer at 10 μM; molar ratio 20:1), the cells were washed with PBS and fixed with 3.7% formaldehyde solution for 1 h. Then the cells were washed 3 times with PBS for 20 min. Hoechst 33342 was utilized to stain DNA (2 h, 1 mg/mL), and F-actin was stained with Texas Red-X Phalloidin (2 h, 0.1 μg/mL). The cells were observed with confocal laser scanning microscopy platform TCS SP8 (Leica Microsystems, Wetzlar, Germany) with an objective of 63×/1.40 (HC PL APO CS2, Leica Microsystems, Wetzlar, Germany) at excitation wavelengths: 405 nm (Hoechst 33342), 488 nm (FITC) and 565 nm (Texas Red-X Phalloidin).

### 2.8. Statistical Analysis

The results are presented as mean ± SD. Data were analyzed by the one-way ANOVA test, followed by Tuckey’s analysis using Origin software; *p* < 0.05 was accepted as statistically significant. All experiments were performed in at least 3 independent replications.

## 3. Results

### 3.1. Live–DeadTest Using Flow Cytometry

For the live–dead test, the MCF-7 cells were exposed to dendrimers or dendrimer/siRNA complexes at different concentrations for 24 h or 48 h and stained with calcein AM (Calc) and propidium iodide (PI). Three main populations can be distinguished after measurements: calcein positive/PI negative (Calc+/PI−), considered live cells, calcein positive/PI positive (Calc+/PI+), considered the cells with disturbed cell membranes (apoptotic cells) and calcein negative/PI positive (Calc−/PI+) cells, considered death cells. The data show that CBD-1 and CBD-2 influenced cell viability in a dose-dependent manner, the higher concentrations of dendrimer caused the number of Calc+/PI− cells to decrease, while the number of Calc+/PI+ and Calc−/PI+ cells increased (Figure 3 and Figure 4).

For cells cultured in 2D exposed to dendrimers for 24 h, a statistically significant decrease of the number of Calc+/PI− cells was observed at 1 μM and 2 μM for CBD-1 (Figure 3). CBD-2 did not lead to any significant decrease of the number of Calc+/PI− cells. This means that CBD-1 seems to be more cytotoxic to the cells cultured in 2D than CBD-2.

In the case of the cells cultured in 3D after 24 h of the treatment, with CBD-1, statistically significant effects for Calc+/PI− cells were visible for dendrimer at 20, 30 and 50 μM and the number of the Calc+/PI− cells was reduced to 86.63 ± 4.41%, 87.43 ± 0.86% and 79.67 ± 4.50%, respectively (Figure 4). For CBD-2, statistically significant effects on Calc+/PI− cells appeared with the dendrimer at 10, 20, 30 and 50 μM, causing a decrease in the number of Calc+/PI− cells to 85.50 ± 2.65%, 67.20 ± 2.26%, 44.78 ± 6.33% and 13.08 ± 4.00%, respectively (Figure 4). This dendrimer was more toxic than CBD-1. Along with the decrease of the number of the Calc+/PI− cells, an increase of the number of Calc+/PI+ cells and Calc−/PI+ cells was noted (Figure 4).

For experiments with dendriplexes, dendrimers at 5 and 10 μM were used, the concentrations at which dendrimers (siRNA carriers) were not toxic to cells. The dendrimer/siRNA molar ratio was 20:1. Statistically significant decreases of the percentage of Calc+/PI− cells compared to NC was visible for complexes CBD-1/Bcl-2, CBD-2/Mcl-1 and CBD-2/Bcl-2, at concentrations of dendrimer at 10 μM (Figure 5). The number of Calc+/PI− cells decreased to approx. 90%. For all complexes with both dendrimers at 5 μM and for CBD-1/Mcl-1 at 10 μM there was no significant effect (Figure 5).

Comparing the decrease in the number of Calc+/PI− cells after incubation with dendriplexes to the number of Calc+/PI− cells after incubation with the dendrimer alone, statistically significant effects were found for CBD-1/Mcl-1 (CBD-1 at 5 μM) vs. CBD-1 (5 μM), for CBD-1/Bcl-2 (CBD-1 at 5 μM) vs. CBD-1 (5 μM) and for CBD-1/Bcl-2 (CBD-1 at 10 μM) (Table 1).

The effect of dendrimers and dendriplexes on the MCF-7 cells cultured in 3D was also measured after 48 h of incubation with dendrimers alone and their complexes with siRNAs. The dendrimer concentration was 10 μM and dendrimer/siRNA molar ratio was 20:1. A statistically significant decrease of the percentage of Calc+/PI− cells compared to NC was recorded for dendrimer CBD-2 and CBD-2/siRNAs (Figure 6), but the number of live cells was still ~90%. No statistically significant effect was apparent for dendriplexes vs. dendrimer for Calc+/PI− cells (for both dendrimers).

### 3.2. Apoptosis Detection by Flow Cytometry (Annexin V-FITC Apoptosis Detection Test)

For apoptosis detection, MCF-7 cells in 3D cultures were exposed to CBD-2 dendrimer or CBD-2/siRNA complexes for 24 h and the Annexin V-FITC Apoptosis Detection Test was performed. Three populations could be distinguished after measurements: FITC+/PI−, apoptotic cells with the changes in phosphatidylserine location in the plasma membrane, FITC+/PI+, apoptotic cells with the changes in phosphatidylserine location in the plasma membrane and simultaneously with the disturbances in the continuity of the plasma membrane, and FITC-/PI+, related to the dead cells.

There was no significant increase in the percentage of FITC+/PI− cells compared to untreated cells. A statistically significant increase in the percentage of FITC+/PI+ was visible for CBD-2/Bcl-2. There was no significant increase in the percentage of FITC-/PI+ cells (Figure 7) and no statistically significant effects for dendriplexes vs. dendrimer alone for FITC+/PI− cells.

### 3.3. Cellular Uptake by Flow Cytometry and Confocal Microscopy

Flow cytometry analysis was used for quantification of dendriplexes taken up by the cells. Analysis was performed using both dendrimers and both siRNAs labeled with FITC. The data (Figure 8) show that in 3.83 ± 0.63% of the cells and in 5.85 ± 0.93% of the cells, complexes with CBD-1 were detected (CBD-1/Mcl-1 and CBD-1/Bcl-2, respectively), whereas for CBD-2, complexes were detected in 6.95 ± 0.93% and in 7.35 ± 1.59% of the cells (for CBD-2/Mcl-1 and CBD-2/Bcl-2, respectively).

Confocal microscopy was used for visualization of internalized complexes inside the MCF-7 cells. Both, Mcl-1 and Bcl-2, complexed with CBD-1 and CBD-2 were visible in the cytoplasm as small green dots (Figure 9).

### 3.4. Live–Dead Test after Treatment with Three Doses of Dendrimers or Dendriplexes Using Flow Cytometry

Since 24 and 48 h of treatment did not cause significant changes in the number of live cells, we performed an experiment in which three doses of dendrimers and dendriplexes were added to the 3D cell culture. Statistically significant effects in reduction in the number of Calc+/PI− cells were visible for all tested groups (CBD-1 and CBD-2 alone and with both tested siRNAs). However, the most visible effect was seen with CBD-1/Mcl-1 and CBD-1/Bcl-2. The number of live cells was reduced to 15.00 ± 9.88% and 8.63 ± 4.92%, respectively. Simultaneously, the number of Calc−/PI+ cells increased to 63.00 ± 9.60% and 71.33 ± 5.71%, respectively (Figure 10).

Comparing the decrease in the number of Calc+/PI− cells after incubation with 3 doses of dendriplexes to the number of Calc+/PI− cells after incubation with 3 doses of dendrimer alone, a statistically significant effect is visible for CBD-1/Mcl-1 vs. CBD-1 and for CBD-1/Bcl-2 vs. CBD-1 (Table 2).

## 4. Discussion

Initially, gene therapy was a therapeutic process in which missing cellular functions were reintroduced by providing a normal copy of a mutated gene into target cells. Currently, protein-coding cDNAs are used in gene therapy to influence cell behavior. These applications include the influence on cell cycle regulatory proteins to block cancer cell proliferation, immune cell activation by transferring gene coding for co-stimulatory proteins into cancer cells, the secretion of cytokines and growth factors coding for neurotrophic factors in Alzheimer’s or Parkinson’s diseases, and the production of angiogenic factors in cardiac or peripheral ischemia. Small nucleic acids, DNAs or RNAs with regulatory functions as well as protein-coding nucleic acids can be used [3]. For example, siRNA can silence target genes by the degradation of complementary mRNA in the cytoplasm [15]. However, siRNA must be transported by carriers, since siRNA alone is repelled from the anionic cell membrane because of its anionic charge and it is not resistant to cellular enzymes [16,25]. Dendrimers are among nanomaterials considered siRNA carriers. They are synthetic polymers, characterized by a branched structure and a large number of terminal groups for interaction with a range of compounds, including siRNA [26,27,28,29,30]. Using dendrimers as nucleic acid carriers is advantageous due to the possibility to control their structure and size, high density of ligand and functionality, and high structural and chemical homogeneity [26].

Transport of siRNA into cells by dendrimers has been investigated by other researchers. For example, Dzmitruk et al. [21], tested three groups of cationic dendrimers: phosphorous based dendrimers, poly(amidoamine) (PAMAM) dendrimers and carbosilane dendrimers, as carriers for anticancer siRNAs (Bcl-xl, Bcl-2, Mcl-1 and a scrambled sequence siRNA). They showed that tested dendrimers complexed with siRNAs could be transported into HeLa and HL-60 cells effectively. Michlewska et al. [13] and Krasheninina et al. [31] proved that various cationic dendrimers could transport siRNA and transfect HeLa and HL-60 cells. Weber et al. [32] proposed amino-terminated carbosilane dendrimers (CBS) as carriers for the siRNA used to reduce HIV replication in peripheral blood mononuclear cells (PBMC) and the lymphocytic cell line SupT1. However, these experiments were conducted using 2D or suspension cell cultures. The physiological environment for cells is 3D and this is crucial for their growth and metabolism [6,22]. For this reason, 3D spheroids are a better model for testing substances (e.g., drugs and nanoparticles), since the results reflect more accurately in vivo cellular responses [6,23]. 2D cultures are deprived of any process of transporting nanoparticles through cell layers and this could affect toxicity results [22]. In addition, some ECM compounds are expressed in spheroids at high levels; thus, spheroids imitate barriers seen in vivo more adequately [33].

The MCF-7 cell line, cultured as spheroids, was chosen for our experiments. We tested two types of cationic carbosilane dendrimers (CBD-1 and CBD-2) in complexes with pro-apoptotic siRNAs (Mcl-1 and Bcl-2), and investigated their influence on MCF-7 cells cultured in 3D. Dendrimers and their complexes with siRNA were characterized and reported previously [14,24].

Firstly, flow cytometry analysis of the cells cultured in 2D, after 24 h incubation with dendrimers was prepared. Those experiments confirmed our previous results [14], that CBD-1 is more toxic for cells in 2D than CBD-2 (Figure 3). Statistically significant decreases of the number of live (Calc+/PI−) cells was visible for CBD-1 at 1 μM. Since the toxic effect on cells is reduced in the spheroid compared to 2D models [22], experiments on 3D cell culture used higher concentrations of dendrimers. Interestingly, the cells cultured as spheroids had a decrease in the number of Calc+/PI− cells higher with CBD-2. Small but statistically significant effects were visible for CBD-2 at 10 μM. Reduction in the numbers of Calc+/PI− was higher with the increasing concentration of CBD-2. For CBD-1, statistically significant decreases in numbers were visible for dendrimers at 20, 30 and 50 μM, but the number of live cells was still above 86%.Therefore, an opposite toxicity behavior was observed between 2D and 3D cultures. For experiments with siRNAs, we chose dendrimers at concentrations that were non-toxic for cells: 5 and 10 μM. Delivery of siRNA using tested dendrimers did not increase the cytotoxic effect in cells either after 24 or 48 h. Measurements of apoptosis did not show an increased level of apoptosis marker after 24 h exposure of spheroids to CBD-2 and its complexes with Mcl-1, but a significant effect was visible for apoptotic cells with a damaged cell membrane after incubation with CBD-2/Bcl-2 (Figure 7).

Measurements of internalization of dendriplexes and microscopy images confirmed that dendriplexes were transported into the cells growing as spheroids. Flow cytometry analysis of internalization indicated that CBD-2 transports siRNAs more effectively than CBD-1, after 24 h of treatment, which could account for the higher cytotoxic effect of CBD-2 on cells in 3D.

Since 24 or 48 h treatments did not lead to changes in viability of the cells, experiments were designed with longer times of exposition to dendrimers and dendriplexes. We added 3 doses of dendrimers or dendriplexes to the cells at 48 h intervals. The results showed a decrease in cell viability, with the greatest effect being visible for CBD-1 complexed with siRNAs. Both dendrimers contain 8 surface cationic groups, which are considered a reason for the cytotoxic action of dendrimers [34]. However, the nature of these moieties is different. CBD-1contains tertiary ammonium groups (RNHMe2Cl), which are dependent on pH, whereas in the CBD-2 structure pH-stable quaternary ammonium groups (RNMe3I) are observed. Another subtle difference is associated with hydrophobicity—the (RNMe3I)-terminated dendrimer is slightly more hydrophobic. Thus, the difference in cytotoxic activity can be explained by different mechanisms of action [14]. However, differences in action are not the only explanation. The differences in cytotoxicity for CBD-1 and CBD-2 and their complexes with siRNAs may result from differences in distribution of nanoparticles within spheroids. It is possible that CBD-2 can penetrate only to the external layers of spheroids. Complexity of the tissue can affect delivery of nanoparticles into the cells. Moreover, the ECM, which is a mixture of proteins and compounds, forms a negatively charged and viscous barrier [33]. The ability of nanoparticles to penetrate through the ECM is affected by their size, charge and surface chemistry. Tchoryk et al. [33] proved that larger particles penetrate less into the core of the spheroid. They investigated poly(styrene) particles of size 30, 50 and 100 nm, observing that within the first 2 h of incubation, over 70% and 80% of the cells in the spheroid internalized particles of 30 and 50 nm size, respectively, and less than 10% of the cells had particles of 100 nm size [33]. Tang et al. [35] showed that camptothecin-silica nanoparticle conjugates of 50 nm were more effectively taken up by mouse-tumor models in vivo and ex vivo than particles of 200 nm. In our studies, confocal images of spheroids suggest that complexes CBD-2/siRNA may aggregate into larger particles and this could be the reason for lower penetration of CBD-2/siRNA complexes and consequently lower toxicity.

Our results suggest that CBD-1 is better as an siRNA carrier as it transports siRNA into the cells and is not cytotoxic for MCF-7 cells cultured as 3D spheroids. Finally, siRNA leads to apoptosis in cancer cells.

Recent research has shown other technologies with potential to correct gene mutations. In 2014, the CRISPR-Cas system was prepared to be used in various organisms to change or silence gene sequences [36]. The CRISPR (Clustered Regularly Interspaced Short Palindromic Repeats) is a region in the prokaryotic genome, consisting of palindromic sequence inserts [36,37]. In this region, Cas protein preserves genetic information. The CRISPR-Cas complex detects foreign genetic information, then stores it and finally destroys it. It works as a antivirus unit in prokaryotic cells [36]. Studies with the use of bacteria and mammalian cells revealed that biologically engineered CRISPR-Cas technology is promising for repairing gene mutations. However, the application of this system has some limitations because of unpredictable side effects and consequences for the next generations [36]. Currently, the CRISPR-Cas system is used in diagnostics to detect RNA and DNA [37].

Besides systems for correction of mutated genes, other anti-tumor approaches are under investigation. Semi-interpenetrating polymer network (semi-IPN) hydrogels containing extracellular polysaccharide Salecan were examined as containers for an anticancer drug doxorubicin (DOX). Hydrogels without a drug were not toxic for HepG2 and A549 cells, while hydrogels with a DOX released a drug into cells, leading to cell death [38].

## 5. Conclusions

Analysis of the internalization of dendriplexes (confocal microscopy and flow cytometry) proved that CBD-1 and CBD-2 complexed with both tested siRNAs (Mcl-1 and Bcl-2) can penetrate into spheroids. Flow cytometry analysis of cytotoxicity of dendrimers alone showed differences in relation to their ability to reduce the viability of MCF-7 cells cultured as spheroids, and CBD-2 seems to be more toxic. The connection of siRNAs to tested dendrimers did not decrease viability of cells when compared to viability after treatment with dendrimers alone in 24 h treatments. However, analysis of cytotoxicity after incubation with 3 doses of dendriplexes revealed the toxic potential of CBD-1 complexed with both siRNAs. This indicates that the effect of dendriplexes on MCF-7 cells cultured as spheroids depends on a variety of factors as a mechanism of action resulting from the structure of the molecule, and also depends on the size of the complex and doses of dendriplexes added at specified time intervals.

Our results on the MCF-7 cells cultured as 3D spheroids suggest that CBD-1 is the better siRNA carrier as it is not cytotoxic and transports siRNA into the cells. Finally, siRNA leads to apoptosis in cancer cells.

It is worth testing other dendrimers at different concentrations, adding doses of dendrimers in different time intervals to choose the best variant.

It is important to test anticancer drugs or materials used in gene therapy on 3D cell cultures, as 3D spheroids mimic tumor structure more properly than monolayers.

## Figures and Tables

**Figure 1 cells-11-01697-f001:**
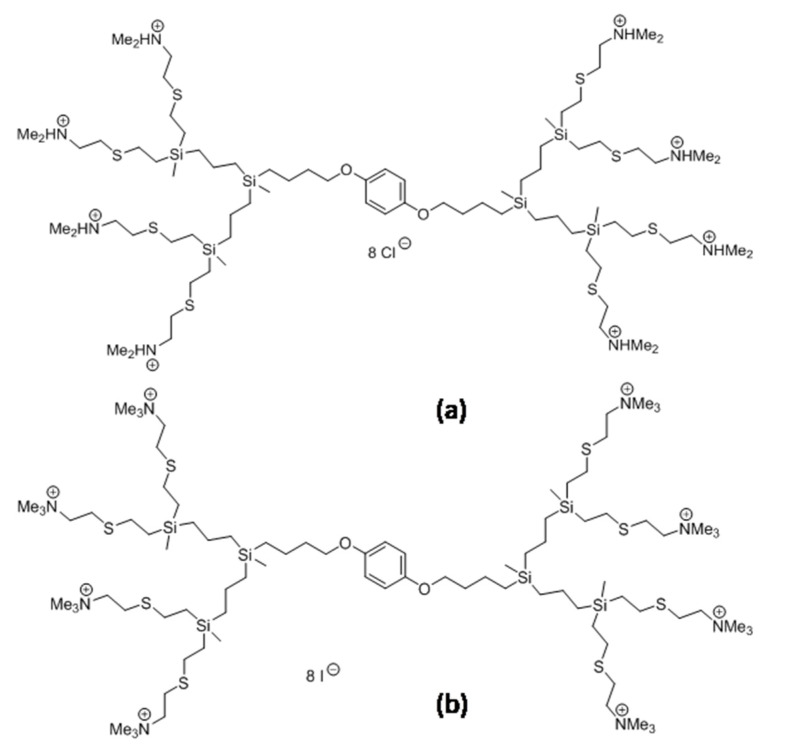
Molecular structure of carbosilane dendrimers: CBD-1 (**a**) and CBD-2 (**b**).

**Figure 2 cells-11-01697-f002:**
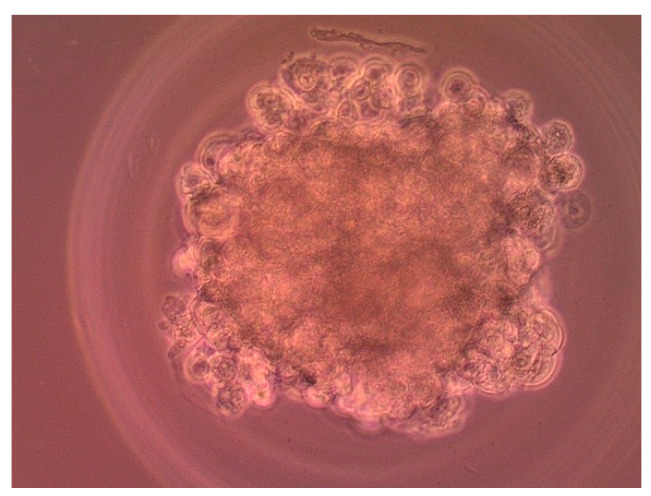
Microscopy image of 3D cultured MCF-7 cells after 7 days after seeding.

**Figure 3 cells-11-01697-f003:**
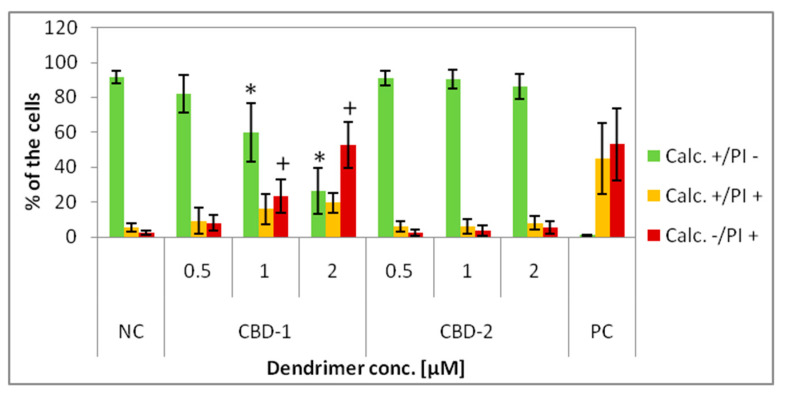
Percentage of 2D cultured cells Calc+/PI−, Calc+/PI+ and Calc−/PI+ after 24 h exposure to dendrimers: CBD-1 and CBD-2 at different concentrations. * *p* < 0.05 vs. NC for group Calc+/PI; + *p* < 0.05 vs. NC for group Calc−/PI+. *n* = 6.

**Figure 4 cells-11-01697-f004:**
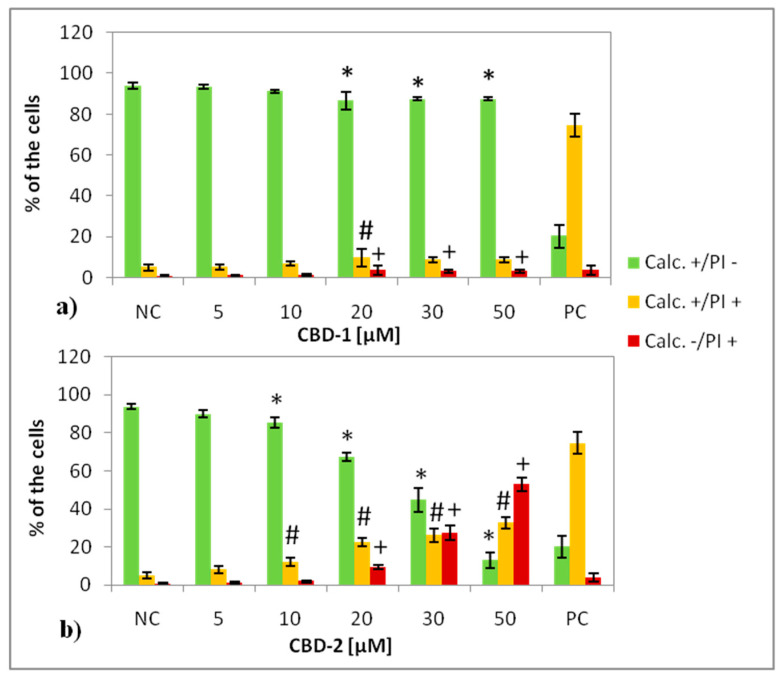
Percentage of 3D cultured cells Calc+/PI−, Calc+/PI+ and Calc−/PI+ after 24 h exposure to dendrimers: CBD-1 (**a**) and CBD-2 (**b**) at different concentrations. * *p* < 0.05 vs. NC for group Calc+/PI−; # *p* < 0.05 vs. NC for group Calc+/PI+; + *p* < 0.05 vs. NC for group Calc−/PI+. *n* = 6.

**Figure 5 cells-11-01697-f005:**
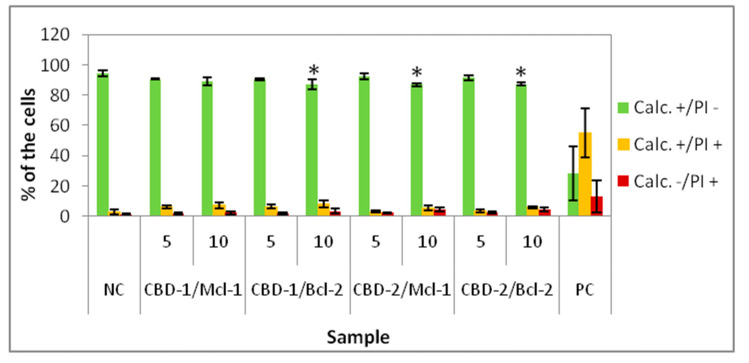
Percentage of 3D cultured cells Calc+/PI−, Calc+/PI+ and Calc−/PI+ after 24 h exposure to dendriplexes. Dendrimers concentrations: 5 and 10 μM; CBD/siRNA molar ratio 20:1. * *p* < 0.05 vs. NC for group Calc+/PI−. *n* = 6.

**Figure 6 cells-11-01697-f006:**
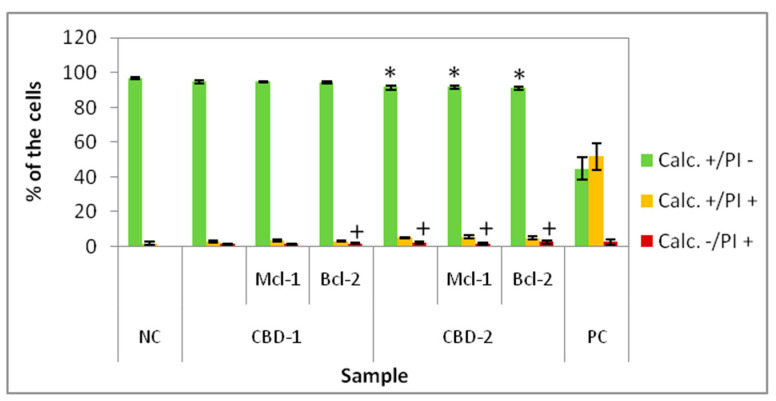
Percentage of 3D cultured cells Calc+/PI−, Calc+/PI+ and Calc−/PI+ after 48 h exposure to dendrimers and dendriplexes. Dendrimers concentration: 10 μM; CBD/siRNA molar ratio 20:1. * *p* < 0.05 vs. NC for group Calc+/PI−; + *p* < 0.05 vs. NC for group Calc−/PI+. *n* = 6.

**Figure 7 cells-11-01697-f007:**
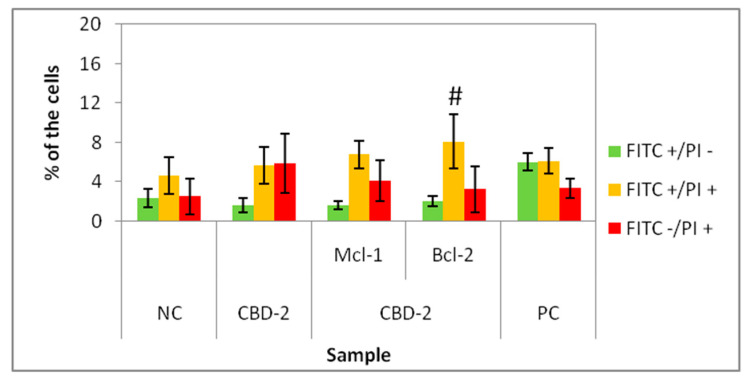
Percentage of 3D cultured cells FITC+/PI−, FITC+/PI+ and FITC-/PI+ after 24 h exposure to CBD-2 and CBD-2/siRNA complexes. Dendrimer concentration: 10 μM; CBD/siRNA molar ratio 20:1. # *p* < 0.05 vs. NC for group FITC+/PI+. *n* = 6.

**Figure 8 cells-11-01697-f008:**
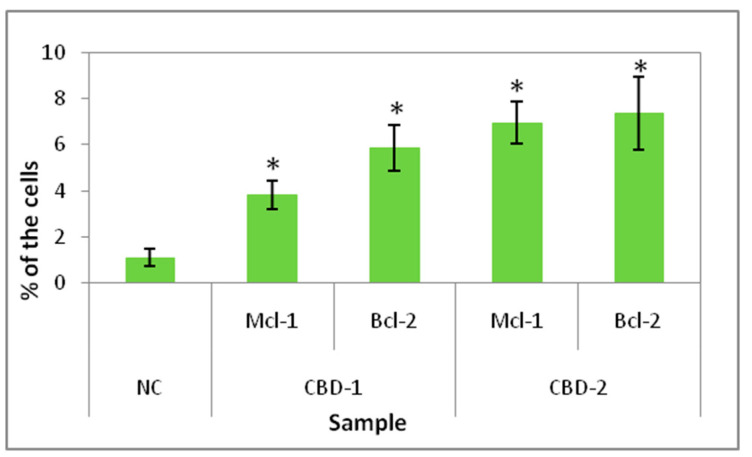
Percentage of 3D cultured cells with complexes of CBD/siRNA-FITC in their cytoplasm after 24 h exposure to dendrimers and dendriplexes. Dendrimer concentration: 10 μM; CBD/siRNA molar ratio 20:1. * *p* < 0.05 vs. NC. *n* = 4.

**Figure 9 cells-11-01697-f009:**
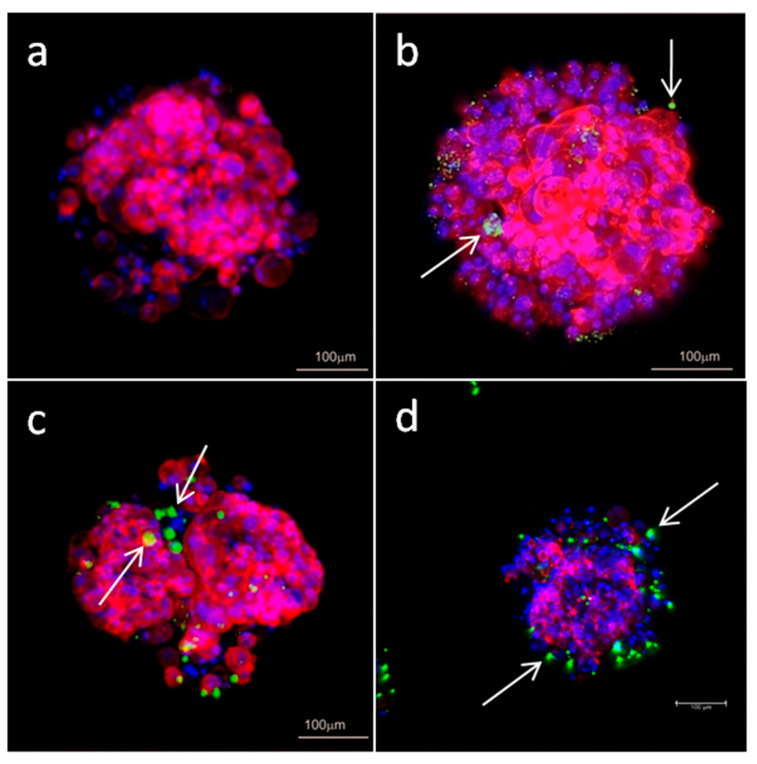
Confocal microscopy images of 3D cultured MCF-7 cells after 24 h incubation with FITC labeled siRNA: (**a**) Mcl-1, (**b**) CBD-1/Mcl-1, (**c**) CBD-2/Mcl-1, (**d**) CBD-2/Bcl-2. The concentration of siRNA was 0.5 µM and the dendrimer/siRNA molar ratio was 20:1. Arrows indicate complexes dendrimer/siRNA.

**Figure 10 cells-11-01697-f010:**
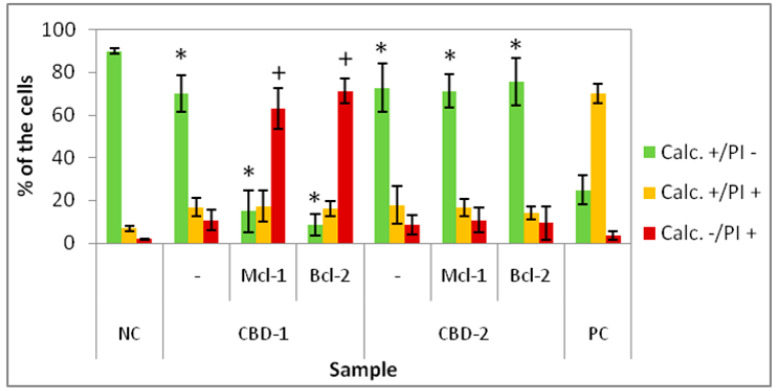
Percentage of 3D cultured cells Calc+/PI-, Calc+/PI+ and Calc-/PI+ after exposure to three doses of dendrimers and dendriplexes. Dendrimers concentration: 10 μM; CBD/siRNA molar ratio 20:1. * *p* < 0.05 vs. NC for group Calc+/PI-; + *p* < 0.05 vs. NC for group Calc-/PI+. *n* = 6.

**Table 1 cells-11-01697-t001:** Statistical significance for % of the Calc+/PI− 3D cultured cells after treatment with dendriplexes vs. % of the Calc+/PI− cells after treatment with dendrimer alone. * *p* < 0.05, -no significance.

CBD 5 μM		CBD 10 μM	
CBD-1/Mcl-1 vs. CBD-1	CBD-1/Bcl-2 vs. CBD-1	CBD-1/Mcl-1 vs. CBD-1	CBD-1/Bcl-2 vs. CBD-1
*	*	-	*
CBD-2/Mcl-1 vs. CBD-2	CBD-2/Bcl-2 vs. CBD-2	CBD-2/Mcl-1 vs. CBD-2	CBD-2/Bcl-2 vs. CBD-2
-	-	-	-

**Table 2 cells-11-01697-t002:** Statistical significance for % of the Calc+/PI− 3D cultured cells after treatment with 3 doses of dendriplexes vs. % of the Calc+/PI− cells after treatment with 3 doses of dendrimer alone. * *p* < 0.05, -no significance.

CBD 10 μM	
CBD-1/Mcl-1 vs. CBD-1	CBD-1/Bcl-2 vs. CBD-1
*	*
CBD-2/Mcl-1 vs. CBD-2	CBD-2/Bcl-2 vs. CBD-2
-	-

## Data Availability

Not applicable.

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
