# Peer review of "Interaction of Cationic Carbosilane Dendrimers and Their siRNA Complexes with MCF-7 Cells Cultured in 3D Spheroids"

_cells, 2022, doi:10.3390/cells11101697_

Round 1
Reviewer 1 Report
Reviewer report on Manuscript Draft ‘Interaction of Cationic Carbosilane Dendrimers and Their siRNA Complexes with MCF-7 Cells Cultured in 3D Spheroids’.
In this research flow cytometry analysis of cytotoxicity of dendrimers alone showed differences in relation to their ability to reduce the viability of MCF-7 cells cultured as spheroids and CBD-2 seems to be more toxic. The connection of siRNAs to tested dendrimers did not decrease viability of cells when compared to viability after treatment with dendrimers alone in 24 h treatments. However, analysis of cytotoxicity after incubation with 3 doses of dendriplexes revealed the toxic potential of CBD-1 complexed with both siRNAs. It indicates that the effect of dendriplexes on MCF-7 cells cultured as spheroids depends on a variety of factors as mechanism of action resulting from the structure of the molecule, and also depends on the size of the complex and doses of dendriplexes added at specified time intervals.
This manuscript is in the scope of journal is rather well written and interestingly addressed. Manuscript contributes to the field of biotechnology. Therefore, the manuscript can be published after some minor improvements:
Last paragraph in Introduction could address the main findings of this manuscript.
The most recent overviews on most recent advances applied in geneterapy, gene assessment and gene determination (Towards application of CRISPR-Cas12a in the design of modern viral DNA detection tools (Review). Journal of Nanobiotechnology 2022, 20, 41. DOI: 10.1186/s12951-022-01246-7 // Advances and Insights in the Diagnosis of Viral Infections (Review). Journal of Nanobiotechnology 2021, 19, 348. // Towards application of CRISPR-Cas12a in the design of modern viral DNA detection tools (Review). Journal of Nanobiotechnology 2022, 20, 41.) could be overviewed and discussed.
English needs some improvements.
Conclusions could be advanced and future trends in Conclusions could be formulated more efficiently and more clearly.
Reviewer 2 Report
Comments to the Author:
- The novelty of this work should be added in the Introduction. For example, why the author choose cationic carbosilane dendrimers for siRNA delivery: advantages over some classical cationic polymers (PEI…).
- Does the activity of siRNA change before and after loading? The authors need to describe this point.
- Advantages over other anti-tumor systems? such as gel delivery systems (10.1039/C6RA10716H).
- There are some formatting errors in the article. In References, does the S in SiRNA need to be capitalized? Please check carefully and use it properly.
- It could be better if a brief comment (challenges and future prospects) is added at the end of the conclusion.
